# Ocean Heat Content in the Iberian-Biscay-Ireland regional seas

Álvaro de Pascual-Collar[1], Roland Aznar[1], Bruno Levier[2], Marcos G.Sotillo[1]

[1]Nologin Consulting, Avda. de Ranillas 1D, 50018 Zaragoza, España.
[2]Mercator Ocean, 8-10, rue Hermès, 31520, Ramonville Saint-Agne, France

*Correspondence to*: Álvaro de Pascual-Collar (alvaro.depascual@nologin.es)

**Abstract.**

The ocean has the largest heat capacity of any single component of the climate system and plays a key dominant role in global heat content changes. To analyse OHC variability in the Iberia-Biscay-Ireland (IBI) region, several Copernicus Marine reanalysis and observational products are used together to provide multi-product estimations of OHC anomalies over the water column (at layers upper 150 m, 700 m, and 2000 m). This work provides a study of spatial and temporal variability of OHC in the Northeast Atlantic region through the analysis of area-averaged time series, trend maps, and trends linked with the main water masses found in the IBI region.

The work states the key role that subsurface water masses play in the OHC trends in the IBI region. The study shows that, although no significant trends are detected for the surface water layers, the intermediate and deep water layers show significant trends (with rates of +0.40 W/m$^2$). However, the high interannual variability of the subsurface water masses masks these trends. Two of the main water masses described in the region (I.e., the Sub-Artic Intermediate Water and Mediterranean Outflow Water) are found to explain more than 50% of the OHC variability. Since the variability of both water masses is linked to the North Atlantic Oscillation, the work shows the mechanisms by which atmospheric forcing is able to affect the subsurface water layers without showing a signal in the surface.

**Short summary.**

The article comprises the analysis of the Ocean Heat Content in the Northeast Atlantic -IBI- region. The variability of Ocean Heat Content is studied, and results are linked with the variability of the main water masses found in the region. Results show how the coupled inter-annual variability of water masses accounts for an important part of the total Ocean Heat Content variability in the region.

## 1. Introduction

It is well established that the main storage (>90%) of extra heat from increasing greenhouse gasses is absorbed by the oceans; consequently, oceans are the dominant source of changes in global warming (Levitus et al., 2005; Levitus et al. 2012; Trenberth et al. 2014; Cheng et al. 2017; von Schuckmann et al. 2020; Gulev et al. 2021). Additionally, the heat stored in oceans has increased during the last decades at basin and global scales (Gulev et al., 2021; Abraham et al., 2013). Therefore, Ocean Heat

Content (OHC) is one of the six Global Climate Indicators recommended by the World Meteorological Organisation for Sustainable Development Goal 13 implementation (WMO, 2017).

Ideally, the estimations of OHC are integrated over the full depth of the ocean, but for limitations related to the observing system, this is typically done from ocean's surface down to a reduced depth (such as 700 m or 2000 m). The choice of these levels is due to the fact before year 2000, temperature measurements were most often taken in the upper 700 m of the water column, but by about 2005 the Argo array had sufficient space-time sampling to yield an improved measure of OHC down to a depth of 2000 m (von Schuckmann et al., 2016; Abraham et al., 2013; Cheng et al., 2022; Cheng et al., 2017).

The Copernicus Marine service provides marine data covering the global ocean and European seas splitting them in six different regions. One of those, the Iberian-Biscay-Ireland (IBI) region, covers the Northeast Atlantic from Canary Islands (26ºN) up to Ireland (55ºN) being limited by the 20ºW meridian and the continental margin. Among the variety of hydrographic processes taking place on the IBI region we could mention the northern limit of the Canary Current, the Upwelling System that influences the Iberian Peninsula and African Coasts, the spreading Mediterranean waters that influences the intermediate hydrographic properties and promotes the formation of long-lasting eddies, and the influence of the Eastern closure of subtropical gyre that brings warm waters from the Gulf Stream influencing the climate of Western Europe. In this region, literature describes a number of water masses but three groups of salinity extrema mainly determine the intermediate hydrographic structure: a subsurface salinity minima connected with Sub-Artic Intermediate Water (SAIW) at potential density anomalies $27.3 < \sigma\theta$ kg/m3, salinity maxima connected with Mediterranean Outflow Water (MOW) at the neutral density $27.25 < \sigma\theta < 27.45$ kg/m3, and salinity minima connected with Labrador Sea Water (LSW) underneath the isopycnal $27.8 > \sigma\theta$ kg/m3 (Talley and McCarney, 1982; Iorga and Lozier, 1999(a); Iorga and Lozier, 1999(b); van Aken, 2000; Prieto et al., 2013; de Pascual-Collar et al., 2019). The density ranges for these water masses show considerable overlap, which allows both diapycnal and isopycnal mixing.

Among the Copernicus Marine Service products are the Ocean Monitoring Indicators (OMIs), which are free downloadable trends and data sets covering the past quarter of a century. These are indicators that allow us to track the vital health signs of the ocean and changes in line with climate change. These OMIs are proposed to cover the blue ocean, e.g., the Mean Volume Transport across sections from Reanalysis OMI (GLOBAL_OMI_WMHE_voltrp); the green ocean, e.g. the Chlorophyll-a time series and trend from Observations Reprocessing OMI (OMI_HEALTH_CHL_GLOBAL_OCEANCOLOUR_trend); and the white ocean, e.g., the Sea Ice Extent OMI (ARCTIC_OMI_SI_extent). Among the multiple OMIs provided by Copernicus Marine, an important family are the indicators focused on evaluating OHC trends and variability. Particularly, the indicator GLOBAL_OMI_OHC_trend (see Table 1 and Table 2 for more details on the product; hereafter this product will be referred as GLO-OMI-trend) provides a global map of trends of OHC integrated in the upper 2000 m. Attending to the spatial variability of trends in the IBI region, this global OMI shows spatial variability of trends, ranging from negative trends of -5 W/m$^2$ northward of the parallel 40º N, up to positive trends of 2.5 W/m$^2$ observed westward of the Gulf of Cadiz. However, this global information should be treated

cautiously when looking at regions close to coastal areas and marked by the combination of on-shelf and open waters, such as

it occurs in the IBI domain.

In the last decades, the upper North Atlantic Ocean experienced a reversal of climatic trends for temperature and salinity. While the period 1990-2004 is characterized by decadal-scale ocean warming, the period 2005-2014 shows a substantial cooling and freshening (González-Pola et al., 2020, Holliday et al., 2020, Somavilla et al. 2009). Such changes are discussed to be linked to ocean internal dynamics and air-sea interactions (Fox-Kemper et al., 2021; Collins et al., 2019; Robson et al

2016), together with changes linked to the connectivity between the North Atlantic Ocean and the Mediterranean Sea (Masina et al., 2022). Previous works show a consistency of regional OHC with this decadal-scale variability in the IBI region (von Schuckmann et al., 2016; von Schuckmann et al., 2018), however in spite of the year-to-year variability, a long-term warming trend of $0.9 \pm 0.4$ W/m2 in the upper 700m of IBI region is also detected. These studies concluded that the positive Earth's energy imbalance dominates the observed regional changes around Europe, but the year-to-year variability in the region

potentially masks the long-term warming trend.

Since the distribution of various water masses is one of the main sources of variability (spatial and temporal) in the IBI region, analysis of the heat stored by them can provide information on OHC trends at regional and local scales. For example, Potter and Lozzier (2004), based on fifty years of hydrographic data, studied the temperature and salinity trends of the MOW, finding positive trends that lead to a heat content gain in the MOW reservoir that overpasses the average gain of the North Atlantic

basin. Nonetheless, the literature on OHC variability of specific water masses in the IBI region is rather limited. Alternatively, since several studies have concentrated efforts on assessing the thermohaline variability of water masses, they may provide some clues as to what can be expected with respect to OHC.

Regarding the SAIW, Leadbetter et al. (2007) studied the temperature variability of the water column in the period 1981-2005 in a section at 36º N, concluding that the variability in the SAIW is consistent with the displacement of neutral density surfaces

driven by changes in surface wind forcing and linked with North Atlantic Oscillation (NAO).

Several studies have focused on the temporal variability of MOW. The analysis of hydrographic properties in the MOW core concluded that changes in MOW properties are not dominated by changes in Mediterranean Sea Water properties (Lozier and Sindlinger, 2009; Bozec et al., 2011). However, the variability of MOW is strongly influenced by year-to-year processes. Some studies described the MOW inter-annual variability as an oscillation of the water tongue shifting the dominant spreading

pathway and interacting with the underlying water masses such as the North Atlantic Deep Water and LSW (Bozec et al., 2011, de Pascual-Collar, 2019), this variability is also correlated with NAO.

The inter-annual variability of LSW has been described in previous works concluding that part of the variability in the water mass can be explained by diapycnal mixing with the overlying MOW (van Aken, 2000) as well as changes in the source regions over the North Atlantic basin (Leadbetter et al., 2007). Additionally, Bozec et al. (2011) studied the distribution of LSW and

MOW from 1950 to 2006 observing a coupling between the spreading areas of booth water masses and the NAO index.

The present work pursues two main objectives: On one hand, the availability of higher resolution specific regional products allows us to refine the study of OHC trends in the IBI region, thus this work explores the spatial and temporal variability of

OHC trends in the region as well as explains its causes in the period 1993-2021, providing a better understanding of the processes controlling such trends. Additionally, since OMIs are simplified indicators that statistically summarize the ocean information, a study of the oceanographic processes behind each indicator is due to have a proper understanding of the changes represented by the OMI. Thus the work analyses the use and interpretation of OHC OMIs on regional scales and its sources of variability and uncertainty.

This paper is organized as follows: Section 2 presents the datasets used, as well as the methodology applied, to compute the OHC in the IBI region. In Sect. 3 the time series of OHC averaged over the whole IBI region are discussed to provide a general view of the regional trends estimated. Section 4 is devoted to show and discuss maps of trends computed in the same way than the ones provided in the GLO-OMI-trend product. Section 5 analyses the vertical profiles of OHC trends studying the variability associated with different water masses. Section 6 summarizes the availability of the data used in this article. Finally, main conclusions are sum up in Sect. 7.

## 2. Data and methods

Following the same methodology than in previous Copernicus Marine Ocean State Report contributions (Lima et al., 2020; Mayer et al., 2021), the estimates of OHC anomalies were computed in IBI region according to the Equation:

$$OHC = \int_{z_1}^{z_2} \rho_0 C_p (T_m - T_{clim}) dz \tag{1}$$

Where $\rho_0$ is the density at a reference depth ($\rho_0$=1020 kg·m$^{-3}$), $C_p$ the specific heat capacity ($C_p$=4181.3 J·kg$^{-1}$·°C$^{-1}$), $z_1$ and $z_2$ the range of depths to compute the total OHC; $T_m$ the monthly average potential temperature at a specific depth; and $T_{clim}$ the climatological potential temperature of the corresponding month and depth.

As can be seen in Equation (1), the OHC anomalies are obtained from integrated differences between the monthly temperature and the climatological one along a vertical profile in the ocean. In the present work the anomalies have been referenced to the monthly climatology computed between 1993 and 2019. Additionally, the OHC is presented for the typical depths of 700 m and 2000 m, but also for the upper 150 m to analyse the OHC variability in the upper layer.

In order to allow the assessing of uncertainties of results, different Copernicus products were used to provide multi-product estimations of OHC, therefore all results were previously computed for a collection of Copernicus products and combined to give an ensemble mean and the standard deviation of the ensemble. Since the objective of this work is the analysis of OHC in the IBI region integrating results from surface down to a maximum of 2000 m depth, this study has included all Copernicus Marine products that deliver gridded data of potential temperature with vertical coverage from surface down to at least 2000 m. As shown in Table 1 and Table 2, four different products meet these stated conditions: two model reanalysis (the global and regional ones: GLO-REA and IBI-REA, respectively) and two observations-based products (CORA and ARMOR). It is worth mentioning that when OHC was computed, the GLO-REA product did not cover the year 2020, so that year was not considered in the mentioned product.

Each used product is supplied for in specific vertical levels with specific thickness, then all vertical integrations were computed
taking in consideration the thickness of each product level. Similarly, since most of the products used are provided with regular
lat/lon grids, the surface of each grid cell depends on the latitude, and the spatial averages (when used in this study) were
computed considering the specific surface of each product grid cell.

This work shows different results of OHC mixing different products with specific spatial resolutions (see Table 2). The OHC
has been computed for all products using Equation (1) on the grid of the distributed products and integrating results (from
surface down to 150 m, 700 m, and 2000 m) using the product layer thicknesses. The presentation of the results as time series
or as maps implies a difference in the way the results are averaged to compute the ensemble mean and spread.
Regarding the time series of OHC, anomalies were calculated referenced to the climatic mean of the period 1993-2019. Then
Regarding the time series of OHC, anomalies were calculated
referenced to the climatic mean of the period 1993-2019. Then the vertical integrations and spatial average were obtained
preserving the service grid of each product. Therefore, the ensemble time series are averaged from time series computed over
the service grid of each product. However, to compute ensemble results that comprises spatial information (i.e., results
presented as maps), each ensemble member must be interpolated to obtain all estimations on the same grid. Since IBI-REA
and GLO-REA products share the same grid, it was considered as the reference grid, thus CORA and ARMOR products were
spatially interpolated. This interpolation was made over the OHC integrated on the respective layer before averaging the
ensemble. The comparison of interpolated and non-interpolated results of each ensemble member did not show relevant
changes on the field structure or mathematical artifacts.

This paper focuses on the analysis of OHC trends in the IBI regions. For this purpose, OHC trends are analysed on different
time periods within the registry. The selection of time periods attends to different criteria such as the detection of trend changes,
comparison of results with other works or selection of periods representative of large-scale patterns. In the case of the analysis
of vertical trend profiles (Section 5), the time record 1993-2021 has been divided into two periods characterised by a
differentiated behaviour of the NAO (NOAA, 2022), considering that (i) the NAO index in 1994
was moderately positive, (ii) it has a minimum in 2010 and (iii) a maximum in 2018; the periods 1993-2010 and 2010-2019
were selected as representative of NAO changes of phase from positive to negative (period 1993-2010) and vice versa, from
negative to positive (period 2010-2019), see Fig. S1.

## 3. Analysis of OHC time series

Figure 1 shows combined results of products IBI-REA, GLO-REA, CORA, and ARMOR (listed in Table 1 and Table 2), with
panels displaying the ensemble mean and the ensemble standard deviation of OHC for the period 1993-2021. Attending the
ensemble spread, time series reveal an increase of uncertainties when deeper layers are included (0-700m and 0-2000m); this
result can be explained by the decrease of observational data available in deeper levels. It is worth noting the decrease of
uncertainties observed after 2003 in the time series integrated over the upper 2000 m. These differences in uncertainties are

explained considering the remarkable improvement in the global ocean observing system achieved with the implementation of the global Argo array in 2005 (von Schuckmann et al., 2016; Abraham et al., 2013; Cheng et al., 2022; Cheng et al., 2017). The period 1999-2001 shows the larger uncertainties observed in the record. This indicates larger discrepancies between ensemble member in representing the minimum of OHC observed in this period. These discrepancies affect both, the year when the minimum is reached (while GLO-PHY-REA estimates the minimum in 2000, the other products reach the minimum in 2001), and the magnitude of the minimum values (agreeing all products in observe negative anomalies during this years).

Trends of time series obtained in Figure 1 show a barely significant warming of IBI region in the upper (0-700 m; $0.39 \pm 0.27$ $W/m^2$) and intermediate-deep layers (0-2000 m; $0.40 \pm 0.39$ $W/m^2$); conversely, such trend does not affect the surface layer, where the trend is not significant. Considering that the trend integrated at 0-2000 metre is at the limit of significance, we can consider this to be a mathematical result, thus results suggest a dominance of the variability at intermediate levels over the OHC trend. It is worth to mention that, despite the positive trends observed in the whole period (1993-2020) of time series integrated down to 700 m and 2000 m, it can be appreciated a change of trends of these time series after the year 2005. Table 3 shows the trend analysis by dividing the time record into two halves: the period 1993-2005 and the period 2005-2021. While the results show a significant warming of integration depths 0-700 and 0-2000 in the period 1993-2005, the period 2005-2021 is characterised by a cooling of both integration layers. This result is consistent with the decadal-scale warming and freshening observed in the North Atlantic Ocean in the period 1990-2020 (González-Pola et al., 2020, Holliday et al., 2020, Somavilla et al. 2009). Additionally, this reversal of trend explains the discrepancies observed between the long-term trends estimated in this work and the ones observed at 0-700 m in von Schuckman et al. (2016) ($0.8\pm0.3$ $Wm^{-2}$) and von Schckman et al. (2018) ($0.9\pm0.4$ $Wm^{-2}$) for the periods 1993-2015 and 1993-2016 respectively. The cited works exclude the last years which have a large influence on the overall trend.

## 4. Analysis of regional trends of OHC

Figure 2 shows trend maps of OHC computed at 0-2000m for the IBI region. The figure includes the computation of trends considering two different periods. The period 2005-2019 was selected to allow comparison of results obtained in this work (Figure 2a) with those obtained with the product GLO-OMI-trend (Figure 2b), computed for the same period and cropped in the figure to only show the study region. On the other hand, the period 1993-2021 was selected to study the spatial distribution of long-term trends in IBI region (Figure 2c). The comparison of results obtained in this work with the GLO-OMI-trend (Figure 2, panels a and b), show a high level of agreement, being differences mostly related with the different resolution. Thus, both figures estimate a negative trend that mainly affects the offshore ocean north of 38ºN and a tongue between 31ºN and 38ºN that shows warming trends.

It worth mentioning that the higher resolutions of the products used on this work, allow computing OHC trends along the Nothwestern European shelf showing a significant warming of the region. This warming may seem low when compared with the values observed in open ocean, but it should be considered that this warming is affected by the shallower depths in the

region. For the sake of brevity, Figure 2 only shows results for the 0-2000m layer, however, as occurred in the time series analysed in the previous section, the observed trends (positive and negative) are subsurface intensified, so the larger is the integration depth, the larger is the observed trend (results not shown). Such intensification of trends suggests that both signals are stronger underneath the upper layer, suggesting that they are more related to the evolution of intermediate and deep water masses than to the year-to-year interaction with the atmosphere.

Examining trends computed for the whole time record (Figure 2c) we conclude that, as estimated by the averaged time series in the previous section, the trend calculated for the whole time record indicates a generalised warming of the IBI area. However, the significance of such warming is smaller, existing regions with no evidence of OHC trend.

As occurred with the analysis of time series, trend maps show a dependence of the results on the time period selected for the analysis (comparing Figures 2a, and 2c). This suggests a strong influence of interannual processes on the observed trends. However, it is worth mentioning that the warming trend of the region around 34ºN latitude is consistent for both analyses.

## 5. Analysis of OHC trends across different water masses

Since it has been observed in the data that OHC trends are accentuated in depth, the following section analyses the vertical profile of trends in the region. However, this analysis cannot be carried out for all products and for all points of the grid. Therefore, it has been decided to study the trend profile obtained from the IBI-REA product for two characteristic subregions centred at 35°N and 48°N (Figure 3a, and 3b). Hereafter these subregions will be referred as 35N, 48N respectively.

The IBI-REA product was selected for this purpose because it assimilates observational (in situ and satellite) data and uses the GLO-REA product as initial and boundary conditions, therefore indirectly incorporating from the parent products information from the other products. On the other hand, the subregions chosen were selected to provide a detailed analysis of the main features described in the previous section. A first analysis has shown that, as seen with vertically integrated data, the resulting trends show a dependency on the selected period. Hence, as explained in Methods (Section 2), in order to provide information on the temporal variability of the water bodies in the IBI area, the trend analysis has been performed by dividing the time period into two periods representative of the positive/negative trend of the NAO index: The period 1993-2010 when the NAO evolves from positive to negative and the period 2010-2019 representing the NAO transition from negative to positive.

The analysis of OHC trend maps for these two periods shows that during the transition to negative NAO phases (Figure 3a), the water masses in the region undergo significant warming over almost the entire IBI area. However, this warming is not significant in part of the southern half of the domain, in the vicinity of the Gulf of Cadiz and the Seine and Horseshoe Abyssal Plains. In contrast, the trend map associated with the NAO transition period towards positive values (Figure 3b), reveals a generalised cooling pattern with a warming trend around the Horseshoe region similar to that observed in Figure 2b. Take into account, however, that figures 2b and 3b are not directly comparable, since temporal coverages differ.

The trends of OHC averaged on the subregions defined at 35N and 48N have been computed at each level to obtain a vertical profile of trends. These trend profiles have been presented in Figure 3 (panels c, d, e, and f) combined with the temperature

and salinity data on a yearly basis. Each profile shown in the θ/S diagrams corresponds to the annual mean temperature and salinity observed on the selected period (1993-2010 or 2010-2021) and averaged over the corresponding study region (35 or 48N). The different markers used correspond to the different OHC trends observed for each depth. Therefore, the shown θ/S diagrams allows to discussing the trends of OHC at each layer linking these results with the different water masses observed in the region.

The θ/S profiles shown in Figure 3 (panels c, d e, and f) are consistent with the water masses that the literature has described in the north-eastern Atlantic: The SAIW characterized by a salinity minima at potential density σθ=27.2 kg/m3, the salinity maxima connected with MOW at the neutral density surface σθ=27.6 kg/m3, and a deeper salinity minima corresponding to LSW underneath the isopycnal σθ=27.8 kg/m3 (Talley and McCarney, 1982; Iorga and Lozier, 1999(a); Iorga and Lozier, 1999(b); van Aken, 2000; Prieto et al., 2013; de Pascual-Collar et al., 2019) however, the core of LSW is only clearly visible

in sub-region 48N where this water mass has a greater presence.

Regarding the OHC variability of SAIW, results in Figure 3 show a similar behaviour of the water mass at 48N and 35N. Thus, at both latitudes, the SAIW shows a warming (cooling) associated with the negative (positive) NAO transition. On the contrary, the OHC trends of MOW shows an inverse behaviour between the sub-regions 35N and 48N. Thus, while in subregion 35N, the MOW experiences a cooling (warming) associated with the negative (positive) NAO transition (periods 1993-2010 and

240 2010-2021 respectively), in subregion 48N the opposite occurs, a warming (cooling) of the MOW associated with the negative (positive) NAO transition is observed. These are consistent with the processes described in literature. Thus, the SAIW trends are coherent with the changes in wind forcing associated with NAO described by Hurrel (1995) and Leadbetter et al. (2007); and the MOW results are consistent with studies that observed a significant anti-correlation between the westward and northward transport of MOW (Bozec et al., 2011) and works that describe the east-west shift of water

mass boundaries in the Horseshoe basin (Pascual-Collar et al., 2019).

Apart of the estimation of density levels defining each water mass (shown in Figure 3), an estimation of the heat trends on each water mass was done for the entire study period 1993-2021 (see results in Table 4). It is observed that, in line with the results shown in Figure 2, the overall OHC trends in both regions are positive. Considering that each water mass has different thickness (SAIW approximately 300 metres and MOW approximately 400-500 metres), we

must conclude that although the total trend shown in Table 4 is of the same magnitude for both water masses, the heat gain per cubic metre is higher for SAIW than for MOW. Additionally, it should be noted the relative influence that both water masses have on the water column trends down to 2000 metres explaining the sum of the two water masses (a layer approximately 700-800 metres thick) 88% (51%) of the OHC trend in 35N (48N), respectively.

**6. Conclusions**

The present work uses several Copernicus Marine products to generate a multi-product OMI of ocean heat content from surface down to a set of depths, over the period 1993-2021. These indicators are able to detect decadal trends of OHC. Specifically,

the IBI region presents a warming of +0.39 W/m$^2$ in the upper 700 meters. However, the study of both time series trends and spatial distribution of trends, show a high sensitivity to the time period selected, therefore the high inter-decadal variability detected makes these trends not very significant. On the other hand, the OMIs integrated up to 150 and 2000 metres does not allow to detect clear significant trends of results, which suggest that OHC variability in the upper 2000 meters may be mainly controlled by intermediate levels.

Although the regional analysis of OHC trends integrated in different depths may provide some clues about the origin of such trends, a finer analysis focusing on the different water masses involved concludes that the vertically integrated trend is the result of different trends (positive and negative) contributing at different layers. The analysis of water masses in the region shows the existence of three water masses (well referenced in the literature): SAIW, MOW and LSW. Since the LSW is found in the depth limit here established (2000 m), we put the focus on studying the temporal variability of the heat stored by the other 2 involved water masses: the SAIW and MOW. The study states that the IBI-REA can simulate the OHC behaviour of SAIW and MOW in a consistent way, being the OHC variability associated with the NAO consistent with the variability described in the literature.

Despite the results show a high relationship between the variability of the NAO and the OHC of SAIW and MOW, both water masses show a differentiated behaviour. While SAIW shows a homogeneous behaviour over the whole IBI area (OHC trends associated to NAO have the same sign in subregions 35N and 48N), MOW shows an anti-correlated behaviour between subregions 35N and 48N (when the trend is positive in 35N, in 48N it is negative and vice versa).

The work states the key role that subsurface water masses play in the OHC trends in the IBI region computing the OHC trends for SAWI and MOW over the period 1993-2021. The results show that, despite high interannual variability, both water masses have experienced significant warming in the study period. Additionally, the relative influence that these two water masses have on the integrated OHC trends down to 2000 metres is remarkable. Thus, it is found that the sum of both water masses accounts for up to 88% of the integrated OHC variability from the surface down to 2000 metres. However, at more northerly latitudes, this relative influence decreases to 50%. The greater presence of LSW in the northern regions of the IBI domain suggests this water mass may share relevance with respect to OHC trends with SAIW and MOW; however, this hypothesis has not been demonstrated in the present work.

**Acknowledgements**

The authors thank to Copernicus Marine Environment Monitoring Service for providing the data for the article. Additionally, the helpful comments of Karina von Schuckmann and the other referees are gratefully acknowledged.

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

525

| Product ref. no. | Product ID / acronym / type | Data access | Documentation<br>QUID: Quality Information Document<br>PUM: Product User Manual |
|---|---|---|---|
| 1 | **IBI_MULTIYEAR_PHY_005_002**<br>(IBI-REA)<br>Numerical models | (EU Copernicus Marine Service Product, 2022a) | **QUID:** Levier et al., 2022<br>**PUM:** Amo-Baladrón et al., 2022 |
| 2 | **GLOBAL_MULTIYEAR_PHY_001_030**<br>(GLO-REA)<br>Numerical models | (EU Copernicus Marine Service Product, 2022b) | **QUID:** Drévillon et al., 2022a<br>**PUM:** Drévillon et al., 2022b |
| 3 | **INSITU_GLO_PHY_TS_OA_MY_013_052**<br>(CORA)<br>In-situ observations | (EU Copernicus Marine Service Product, 2022c) | **QUID:** Szekely, 2022a<br>**PUM:** Szekely, 2022b |
| 4 | **MULTIOBS_GLO_PHY_TSUV_3D_MYNRT_015_012**<br>(ARMOR)<br>In-situ observations · Satellite observations | (EU Copernicus Marine Service Product, 2021a) | **QUID:** Greiner, 2021<br>**PUM:** Guinehut, 2021 |
| 5 | **GLOBAL_OMI_OHC_trend**<br>(GLO-OMI-trend)<br>Numerical models · In-situ observations · Satellite observations | (EU Copernicus Marine Service Product, 2021b) | **QUID:** von Schuckman et al., 2021<br>**PUM:** Monier, 2021 |

**Table 1: List of Copernicus Marine products used for the computation of Ocean Heat Content (OHC) in Iberia-Biscay-Ireland region (IBI).**

| Product Reference | Resolution | Temporal Coverage | Vertical Coverage |
|---|---|---|---|
| 1. IBI-REA | 0.083° x 0.083° | 1993-2020 | 0 - 5500m (50 levels) |
| 2. GLO-REA | 0.083° x 0.083° | 1993-2020 | 0-5500m (50 levels) |
| 3. CORA | ~0.36° x 0.5° | 1993-2021 | 0–2000m (152 levels) |
| 4. ARMOR | 0.25° x 0.25° | 1993-2021 | 0–5500m (50 levels) |
| 5. GLO-OMI-trend | 0.25° x 0.25° | 2005-2009 | Integrated 0-2000 (1 level) |

**Table 2~~1~~:** Spatial and temporal coverage of products~~List of Copernicus Marine products used for the computation of Ocean Heat Content (OHC) in Iberia-Biscay-Ireland region (IBI).~~.

535

| Integration Depth (m) | OHC Trend [W/m²] Computed in Period | | |
|---|---|---|---|
| | 1993-2005 | 2005-2021 | 1993-2021 |
| 0-150 | 0.36 ± 0.46 | 0.03 ± 0.25 | 0.10 ± 0.12 |
| 0-700 | **1.6 ± 0.57** | **-0.56 ± 0.35** | **0.39 ± 0.27** |
| 0-2000 | **1.86 ± 1.09** | **-1.02 ± 0.48** | **0.40 ± 0.39** |

**Table 32: Mean OHC trends (in W/m²) averaged in the IBI domain and integrated from the ocean's surface down to 150 m, 700 m, and 2000 m. Confidence interval of trends is computed with 95% confidence. In bold the values where the absolute trend exceeds the confidence interval.**

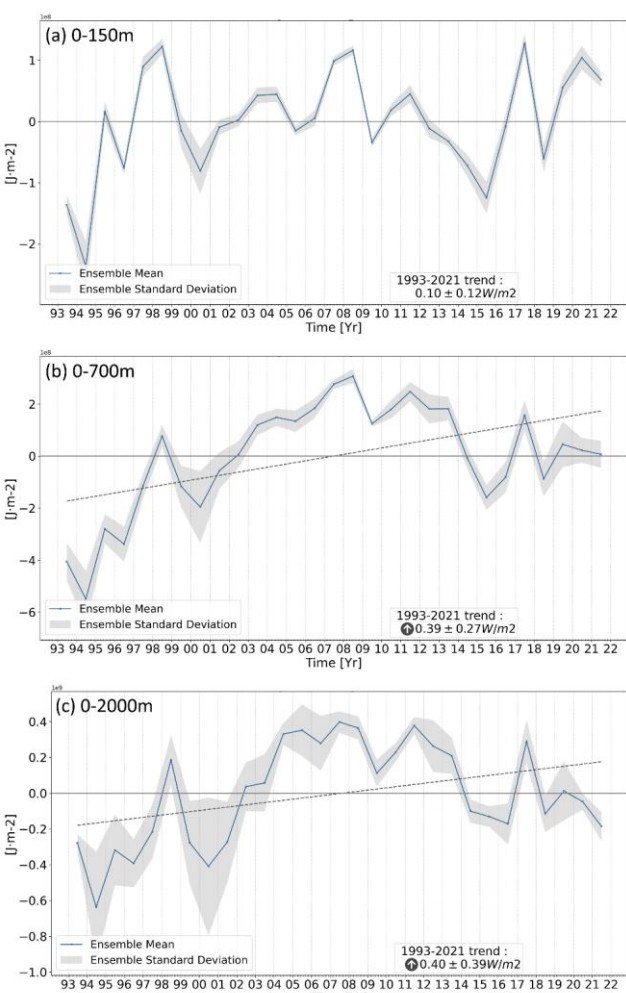

540

**Figure 1: OHC averaged in the IBI domain integrated from the ocean's surface down to 150 m (a), 700 m (b), and 2000 m (c), respectively. Time_series computed from 4 Copernicus Marine products (i.e.: IBI-REA, GLO-REA, CORA, and ARMOR), providing a multi-product approach. Blue line represents the ensemble mean and shaded grey areas represent the standard deviation of the ensemble. The analysis of trends (at 95% confidence interval) computed in the period 1993-2020 is included (bottom-right box). Trend lines (dashed line) are only included in the figures when a significant trend is obtained.**

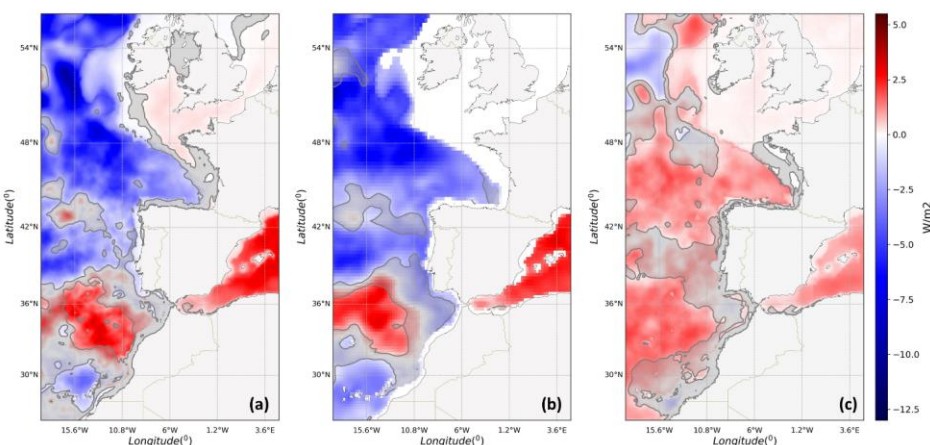

**Figure 2: Regional maps of OHC trend computed at 0-2000 m. (a) over the period 2005-2019 and (c) 1993-2021. Trends computed using the 4 Copernicus Marine products (IBI-REA, GLO-REA, CORA, and ARMOR) providing a multi-product approach. Shaded colours represent mean trends (using all products), while shaded grey indicates areas with less robust signatures (where the noise (ensemble standard deviation of trends) exceeds the signal (ensemble mean))). (b) same as panels a and c but obtained from the product GLO-OMI-trend over the period 2005-2019.**

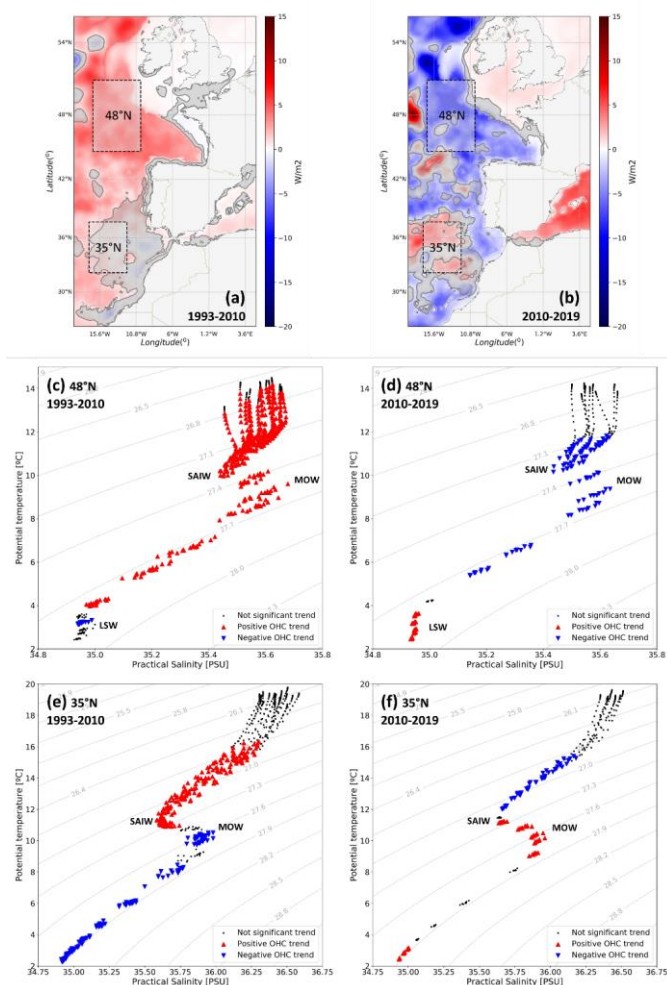

**Figure 3: (a) and (b) maps of regional trends of OHC (0-2000 m) over the period 1993-2010 (a) and 2010-2019 (b) derived from the IBI-REA product. Grey shaded areas represent regions where the trend is not significant (95% confidence). Dashed rectangles denote the subregions 35°N and 48°N where θ/S diagrams are averaged in other panels. (c, d, e, and f) θ/S diagrams averaged in subregions (35°N and 48°N) and over the periods 1993-2010 and 2010-2019. (c) region 48°N period 1993-2010, (d) region 48°N period 2010-2019 (e) region 35°N period 1993-2010, and (f) region 35N° period 2010-2019.**

570

| | Averaged Trend [W/m2] in IBI Subregions. (1993-2021) | |
| --- | --- | --- |
| **Water type** | **35N** | **48N** |
| **Sub-Artic Intermediate Water (SAIW)** | +0.52 (44%) | +0.40 (27%) |
| **Mediterranean Outflow Water (MOW)** | +0.52 (44%) | +0.36 (24%) |
| **Total (0-2000 m)** | +1.18 | +1.46 |

**Table 4:3 Mean thermic trends (in W/m$^2$) of SAIW, MOW, and 0-2000 m in the IBI subregions (35N and 48N) computed for the period 1993-2021 with product IBI-REA. Significance of trends computed with 95% confidence.**

575

