# Peer review of "Ocean Heat Content in the Iberian-Biscay-Ireland regional seas"

_State of the Planet, 2022_

## Referee Comment (RC1)

**Review of "Ocean Heat Content in the Iberian-Biscay-Ireland regional seas" by Pascual-Collar *et al* (2022) for 7th edition of the Copernicus Marine Service Ocean State Report (OSR 7)**

**Summary**

This work reports on Ocean Heat Content (OHC) trends over the Iberian-Biscay-Ireland (IBI) region based on Copernicus Marine products. Averaged over the entire region, OHC from 0 to 2000m shows a positive trend of 0.5 ± 0.4 W/m$^2$. Regional values display an interesting dipole pattern, with negative trends affecting the offshore ocean north of 38˚N, and a positive 'tongue' between 31˚N and 38˚N. The authors then link the OHC trends with the main water masses in the region, revealing how mainly the Mediterranean Outflow Water impacts the region. As OHC is considered an Ocean Monitoring indicator, linking OHC with water masses is a relevant issue, as it brings insights in the mechanisms driving OHC changes. Hence, the preprint is of scientific value and in the scope of the Ocean State Report. However, there is a divergence between the values reported in Figure 1 and the ones reported in the text. Depending on which value is correct, the interpretation of the results can change a lot, and so the conclusions found in the text. With this issue being solved, and the general remarks below being addressed, I believe this would be a good addition to the Report.

**General comments:**

1. As mentioned in the summary, there is a divergence between the values reported in Figure 1 and the ones reported in the text: Figure 1 reports a trend of 0.4 ± 0.3 of OHC integrated from surface to 150m and of 0.1 ± 0.1 from surface to 700m, while in the text the authors report the 0.4 ± 0.3 being from 0-700m.
2. Structure: The structure of the report could be improved. There are methodological details in the results that belong in Data & Methods Section, and results in the Conclusion (which will be indicated in the specific comments).
3. Data & Methods: The methods section could also benefit from explaining how trends were compute (e.g., ordinary least-squares, auto-regressive models, mann-kendall, …), and by giving some brief details about the data sets used in the analysis. Why were only these 4 datasets used? There are other global products besides GLO-REA, CORA and ARMOR which could have been used (for example, CGLOR and ORAS-5 (GLOBAL_REANALYSIS_PHY_001_031)), why were these ones selected? It is also important to mention if the same source of temperature and salinity data are included in these products, since you present an ensemble mean in the results. It would also be relevant to mention here that GLO-REA is used as boundary condition for IBI-REA (and the possible implications of this for the ensemble mean).
4. Discussion: There is a lack of discussion with previous works. For example, in the introduction it is mentioned that previous reports "detected warmings of 0.9±0.4 W/m$^2$ in the upper 700m in the study region". The present study reports a trend of 0.1±0.1 (or 0.4±0.3, depending on which value is correct). The divergence of values should be discussed in the manuscript: how come such smaller trends have been detected now?

5. Region map: This is more of a suggestion, but I think the Introduction could benefit from a map showing the IBI region and indicating the main water masses which are discussed in the text. This would make the text of the introduction read easier, and the interpretation of the results.

**Specific comments:**

- L10: I would change 'model' to 'reanalysis', or say 'model reanalysis' (as in L106).
- The abstract and short summary mention that the work provides an "exhaustive analysis", which made me expect either a lot of data sets included in the analysis, or a lot of results. I think the world "thorough" or another one would be better suited here.
- L39: Could you add some examples of Ocean Monitoring Indicators?
- L42: It is said that the product "GLOBAL_OMI_OHC_trend" will hereafter be referred as GLO-OMI-trend, but in the next sections it is referred mainly as "GLOBAL_OMI_OHC_trend" (for example in paragraph 2 of section 4).
- L49: for which period is the trend of 0.9 ± 0.4 W/m$^2$ from? Can you also add reference to the works that reported such trend?
- The paragraph about water masses (L60-67) in the study region should come before the paragraph describing how previous studies linked these water masses with OHC (L49-59). This would be a good place to actually introduce the study region.
- L68-70: Are these results also for the IBI region? This should be mentioned. And I guess this example belongs to the paragraph from L49-59.
- L72: You say MOW does not dominate the changes in the Mediterranean Sea Water… could you mention what does then?
- L81-83: Here the objective of the work is described. I think should also mention for which periods OHC trends will be analyzed.
- L102: You mention "a collection of data sources", do you mean several Copernicus products (such as CORA, Glorys, …) or several sources of temperature and salinity data (such as Argo gloats, CTDs, XBTs, …)? It was not clear for me. Also "several data sources" (L101) refers to which? If it's to the products in table 4, I don't think 4 products can be classified as 'several'.
- L114: Here should be mentioned which grid resolution was selected for the ensemble mean.
- L116-119: this should be in the methods section.
- Analysis of OHC timeseries: I was wondering if instead of only integrating the values from the surface until the reference depth (150, 700 and 2000m), the analysis could benefit from integrating in intervals? From 0-150m, 150-700m, 700-2000m. This way the actual contribution from each layer would be clearer. With always integrating from the surface, the contribution and uncertainties become cumulative, and not independent, so the results from 0-2000m should reflect the behavior from 0-700m, in addition to the contribution of 700-2000m.
- L123-L127: not only this change in the uncertainties is 'remarkable'. I also found very interesting how the uncertainties are much wider from 99-2002. Could you comment on what could be the cause of that? Would it be related to climate fluctuations, or also to the quality (and quantity) of data?
- L128-L131 & Figure 1: Here is where the results of Figure 1 and the text do not match. The text says the trend of 0.4±0.2 is for the upper 700m, which are in the figure the

numbers for the upper 150m. Then it is concluded that "such a trend does not affect the mixing layer. This result suggests a dominance of the variability at intermediate-deep-levels over the OHC trend". Considering the mixing layer is within the upper 150m (as stated in L100, but could actually be reinforced here). If the numbers in Figure 1 are correct, then I would interpret as: the upper 150m have a positive trend; the upper 700m have an insignificant trend, dominating the variability of OHC, instead of the trend; the overall integrated OHC in the upper 2000m is a positive trend.

- L131: "It can be appreciated a change of trend after the year 2006". What does this mean exactly? Do you mean that if we would analyse trends from 1993-2006, then we would see a negative one and a positive one for 2006-2020? Or do you mean an acceleration in the rate…? Could this be quantified (by a breakpoint analysis for example)?

- L135-138: this should be in the methods section.

- L138: The IBI-REA grid was used as a reference, which is the one with the higher spatial resolution. This mean that CORA and ARMOR had to be downscaled to match this grid resolution, meaning than that each grid cell is not independent anymore (1 value was split into more). Could you comment on effect of this for your analysis?  Would your results change if instead of matching IBI-REA resolution, you would match the data set with coarser resolution?

- L144: what do you mean by "inhomogeneity of uncertainties"?

- L148-152:  You talk here about some of the divergencies of the ensemble and the OMI product. Might also be worth mentioning that positive trends are seen in the ENS in along the Northwestern European shelf, but this is not seen in the OMI product.

- Another thing that caught my eye in Figure 2 is between 30-36˚N the trends are not significant for 0-150 and 0-700m, but they are from 0-2000m. Suggesting that the uncertainties become smaller when the deeper layers are added, which seems counterintuitively to me. Or maybe is just an artifact of the intensification of the trends with the deeper layers (as stated in L155)? Could you comment on that?

- L162-L165: this should be in the methods section.

- L175: It was unclear to me what was meant by 'yearly basis'. Did you use yearly values for the diagrams? Looking at Figure 3 and the results, I believe you used trends of temperature and salinity for the diagrams... but this was unclear.

- Figure 3: Just a suggestion: since you discuss the water masses, maybe you could actually add boxes in the T/S diagrams indicating where is each water mass? (I had to go back and forth between the definition of the water masses and the figures to know which one was at the surface and which one was the intermediate and the bottom one.

- L185: "a positive trend of temperature entails a positive trend of salinity". Is it only temperature influencing salinity, or vice-versa?

- L198: you state that the positive trends in LSW could be explained by the connection between the MOV and LSW. But what about box 49N, in which MOW has negative trends? Then the positive trends seen in LSW in that box could not be due to the MOW influence…

- L209-210 and L218: these conclusions are wrong if the values in Figure 1 of 0-150m are correct.

- L220-228: This is more of results than conclusion. Maybe would fit better in the previous section.

**Technical corrections:**

- L44: Here is the first time the acronym "IBI" is used in the main text. You should write it out Iberian-Biscay-Ireland the first time it is mentioned.
- L45: Typo: 'info' -> information
- L46: Typo: 'regions closed to coastal areas -> close
- L49: OSRs is mentioned for the first time, without being defined (and appears written out in L91). I suggest just using the expanded version in both places, since the texts already has a lot of acronyms.
- L54 and L63 both define Mediterranean Outflow Water (MOW). An acronym is usually only defined once.
- L57-59: You should either use 'although' or 'nevertheless', but not both in the same sentence. If you keep 'although', then the semi-column after limited should be replace by a period.  And in the last part of the sentence "these studies give some…" you can remove "these studies" and change "give" to "giving".
-  L80:  NAO hasn't been defined. And it actually appears written out in L218.Again, I suggest just using the expanded version in both places, since the texts already has a lot of acronyms.
- L103-105: "Since the objective of this work is the analysis of OHC in the IBI region integrating results from surface down to a maximum of 2000 m depth., Tthis study has included all Copernicus Marine products that provide gridded data of potential temperature with vertical coverage from surface down to at least 2000 m. '
- L146: typo: booth -> both
- L155: It should be 'larger' instead of 'bigger': "the larger the integration depth, the larger is the observed trend".
- L208: "being these indicators useful …" -> "these indicators are useful …"
- L210: "Despite the regional analysis …" -> "While the regional analysis …"
- L213: end of the line, between OHC and Table 2 should be a period and not a comma.

---

## Author Comment (AC1)

*Dear Editor.*

*I have read the manuscript entitled Ocean Heat Content in the Iberian-Biscay-Ireland regional seas by Pascual-Collar et.al*

*The manuscript makes use of different Copernicus products that provide, or allows computing, Ocean Heat Content in the IBI region. The work describes the spatio-temporal characteristics and also in terms of depth level integration of OHC series obtained in relationship to water masses in the region.*

*While I do not appreciate inconsistencies in the analysis, I feel that at this stage the manuscript does not present enough new and relevant research to merit a paper on its own. The analysis consists of computing linear trends from a number of available Copernicus products, and the discussion on the outcomes appears to me too descriptive and sketchy. I think that further work is needed in order to improve the manuscript. Some proposals are indicated below.*

First, the authors would like to thank the reviewer for their valuable comments. Addressing his remarks and following his suggestions, the revised manuscript will be certainly improved.

The authors agree to a certain extent with the limitations pointed by the reviewer with respect to the original manuscript. Somehow the OSR structure (together with its limitation in extension and figures) conditions the contribution, and the paper may result a bit sketchy. The authors points the difficulties to compress the information needed to support a full paper in such a brief contribution. In our opinion, the reviewer has evaluated this contribution using considerations of a full paper (and we thank for that), but probably it should be evaluated keeping in mind the mentioned extent contribution limitation associated to the OSR guidelines.

The revised manuscript will be significantly improved following reviewer's suggestion and generalities, pointed by the reviewer, will be reformulated or suppressed.

The authors would like to remark here some scientific achievements of this contribution:

1. The main purpose of the contribution, focused on the analysis of Copernicus OHC OMI on regional scales and its sources of variability and uncertainty, is fully accomplished.
2. This work also provides information of the key role that the variability of subsurface water masses plays in the OHC trends in the IBI region
3. The work provides numeric estimations of warming/cooling for the whole IBI region as well as for specific subregions and water masses.

These estimations are computed using state-of-the-art methods having an intrinsic value by itself, moreover they can be used in future works to compare with other studies.

4. We agree that some discussions in this work left a path open for further investigations, but that is in fact a scientific result usually included in many scientific works.

After a carefully read of the reviewer's comments, the authors can partly agree with them, and we think they can be considered to improve the manuscript. Therefore, we propose to include the following mayor changes in the manuscript that mainly affect sections 4 and 5:

1. Include a better, and clearer, description of the main objective of the contribution: the evaluation of the proposed Ocean Monitoring Indicator in the IBI region.
2. Give a better discussion on the outcomes, considering and following the reviewer's suggestions. analyse the consistency of results and the relevance of interannual variability on OHC trends. To this aim, the authors will deeply modify the Section 4 (Analysis of regional trends) and 5 (Analysis of OHC trends across different water masses).
3. Highlight, as part of the conclusions, the scientific contribution of the work.

*My first concern regards the purpose and value of using (and averaging) 5 different products. I understand that this work is not focused on product intercomparison and detailed documentation of each product is linked on table 1. I expect all products should yield similar outcomes since the bulk of baseline data comes from available ship-based hydrography and Argo floats, however further details on differences between products and especially on the causes of these differences will add value. If all products are very similar there is no point in averaging all five available, otherwise it would be interesting a discussion on which product may suit better. I am in particular confused about differences between the two reanalysis IBI-REA and GLO-REA since both have same resolution and coverage.*

The methodology used in this contribution follows analogous methods to those previously used and discussed in the literature (specifically in the OSR: von Schuckmann et al., 2016; von Schuckmann et al., 2018; Lima et al., 2020; Mayer et al., 2021). Regarding the averaging of 5 Copernicus products, it should be mentioned that we are not as interested in the average itself than in the spread (the differences) of the products. Through the computation of the spread we obtain a proxy for the uncertainties of the indicator and thus, information on the indicator reliability (Lines 101-104). For example, the significant decrease identified in the spread after 2003 (Figure 1c) is a useful information for the user, and a warning on the existence of bigger uncertainties of this indicator in its earliest period 1993-2003.

The discussion of differences observed between products is mainly confronted in Section 3 (Lines 122-127), in this section the authors state than the main differences between products are due to the lack of observational data before the implementation of the Argo array. However, we are open to modify the text in case of the reviewer consider such explanation is not clear enough or some information is missing.

The discussion of which product may suit better is completely out of the purpose of this contribution (indeed, providing such direct product comparisons never was the spirit or goal of the OSR), anyway if that would be the objective, Figure 1 would show a different coloured line for each product. In this work, we assume the ensemble approach, so that, the use of an average of products (even if they are highly correlated) is always better than the use of just one product; and the spread of the members can be used as an indicator of uncertainties.

The authors do not understand where in the text the reviewer observes "differences between the two reanalysis IBI-REA and GLO-REA" because such comparison is not shown in the manuscript. We assume that this conclusion could be related with the results shown in Figure 2 and discussed in Section 4 (Lines 148-151). However, as it is explained in the text and Table 1, the resolution of GLO-OMI-trend is lower (0.25 degrees) than the resolution of our results (0.083 degrees). If this is the source of confusion, we are open to modify the Section 4 to provide a clearer understanding to readers.

The authors consider very fruitful the discussion of any result derived from our work, and we are fully open to include new related information in the resulting contribution. However, we would need further details about where in the manuscript the reviewer observes differences between IBI-REA and GLO-REA.

*My second concern is that the discussion on OHC changes in relationship to water masses is too brief. Changes are interpreted in terms of boundaries advance/retreat, while no definition on boundaries is provided nor are insights on circulations changes that may cause such boundaries shits. I elaborate further on the specific comments.*

Again, we crash here with the size limitations of this OSR paper. We would like to show in this work an analysis as detailed as in, for example, Pascual-Collar et al. (2019). But the issue is that whereas Pascual-Collar et al. (2019) has 17 pages (and 10 figures), the present contribution in review will barely have 7 pages (and 4 figures).

Therefore, this work assumes conclusions derived from other works such as Leadbetter et al. (2007), Bozec et al. (2011), and Pascual-Collar et al. (2019), being these works cited to avoid long explanations. The cited papers develop

the hypothesis of the oscillatory processes of subsurface water masses in the Northeast Atlantic. Additionally, they provide a detailed definition of boundaries, discussion of circulation changes, and relationship with NAO index. This hypothesis could be revised (and improved) on the basis of new research; but as far as we know, currently there is no evidence (i.e. scientific publication) that supports other alternative hypothesis. Therefore, the scientific method supports the use of the current hypothesis to explain the current observations. The authors would be happy to revise the results in case of new scientific information appears on this topic.

According to our understanding, the scope of the OSR is to *"provide a comprehensive and state-of-the art assessment of the state of the global ocean and European regional seas for the ocean scientific community as well as for policy and decision-makers".* On this framework, the authors consider more adequate to present (i) an analysis of the proposed Ocean Monitoring Indicator, (ii) a discussion of the observed trends, and (iii) a discussion of how results are consistent (or not) with the ones seen in previous works; than a deep analysis of the oceanographic processes behind the oscillatory processes of subsurface waters in the Northeast Atlantic.

However, we understand that the discussion of OHC in relationship to water masses can be improved by addressing deep changes in section five. On this regard, we propose to analyse the OHC profiles in two different periods: 1993-2010 and 2010-2018. These two periods are proposed to represent two different behaviours of NAO index: negative trend in the period 1993-2010 and positive trend in the period 2010-2018. We consider this analysis can reinforce our conclusions highlighting the concordance of results with the hypothesis developed on Leadbetter et al. (2007), Bozec et al. (2011), and Pascual-Collar et al. (2019).

--

*Specific Comments*

*l.8 (abstract). The statement that OHC has increased not only globally but at regional scales is almost self-evident; OHC cannot increase globally if it does not also increase in [many/most] regions.*

This assertion will be modified as suggested to avoid redundancies.

*l.10 "observed derived products" sounds weird to me.*

The sentence will be modified as follows:

*"...several Copernicus Marine reanalysis and **observational** products are used together to provide multi-product estimations of OHC anomalies over the water column..."*

*l.15. There is no contradiction between (1) having significant warming and (2) having OHC variability dominated by thermohaline variability of subsurface waters. Indeed, it is neither 'thermohaline variability' dominating 'OHC variability' nor the other way around, both are equivalent. I guess authors are trying to convey that interannual variability due to advective patterns dominate the thermohaline variability/OHC. This should be made clearer across the ms. This is said again in the short summary.*

The authors agree on this point, and the text will be modified to clarify the role of advective patterns and water mass distribution. This sentence is in a section that will be strongly changed in the revised manuscript.

*l.100. It is indicated that the level 150m is chosen to 'analyse the OCH varibility in the mixing layer', however mixed layer depth varies strongly across the IBI domain from several hundredths of meters west of Ireland to tenths in the south (e.g. http://mixedlayer.ucsd.edu/). If authors wish to analyze OHC variability/trends in the mixing layer they should not use a fixed 150 m reference depth.*

The authors are aware that in oceanography is quite usual to accept the depth around 100m or 150m as an average depth to represent the mixing layer. However, in order to prevent misunderstandings, it was avoided the use of "mixing layer" in relation with the OHC integrated from surface down to 150 m depth, being replaced by "surface layer" or "upper layer".

*l.128. ss. the authors highlight the increase of OHC for the whole period 1993-2020 but in the next section it is decided to compute trends for the period 2005-2019 since the global Argo array become dense enough, and this period shows a cooling.*

The authors fully agree with the reviewer on this point, we consider this contradiction as something that blurs the conclusions.

We propose to modify section 4 showing the regional trends not only for the period 2005-2019 but also for the whole period 1993-2020. We consider the difference of results computed in these two periods an important result worthy to be discussed in the text.

Additionally, Figure 1 is modified accordingly, thus it will show trends computed in the two periods (2005-2019 and 1993-2020). The text will be consequently adapted to discuss this result.

*l.184. the cooling trends in Fig.3 are computed for the period 2005-2019, so the statement that this is consistent with changes in thermocline thickness described in*

*2007 by Leadbetter et.al. (right at the start of the series or even before) deserves further explanation. Is it that the process described by Leadbetter initiated cooling trends in the IBI region? Is it suggested that this mechanism continued operating the following decade?. The authors should notice the large scale North Atlantic freshening/cooling well documented for the 2010s (e.g. https://doi.org/10.1038/s41467-020-14474-y https://doi.org/10.17895/ices.pub.7537)*

We understand the comment of the reviewer. This sentence is too short and summarized, and a clearer discussion of results regarding Leadbetter et al. is due. As explained in the respond to the general comments, deep changes will be done in this section to provide a better interpretation of results.

Additionally, we thank the reviewer's recommendation to enhance the bibliography by means of the inclusion of the ICES report. The detailed information of such report on the North Atlantic helps in the discussion of results.

*l.189. the discussion on the displacement of the MOW boundaries in the Horseshoe basin requires further explanation. I do not see clear relationship between displacements of the 'MOW boundaries' westward and warming/salt increase in the region (besides, boundaries are not defined). As long as source waters properties do not vary, a westward shift of the boundary should only cause warming west of the original boundary placement. The warming/salt-increase in this sub-basin makes me feel that the waters are slowed/retained. If so, possible reasons should be discussed.*

As we explained in the general comments, this contribution assumes the conclusions derived from previous works, especially Leadbetter et al. (2007), Bozec et al. (2011), and Pascual-Collar et al. (2019), considering these works provide solid results that allows to accept their hypothesis. The discussion of such hypothesis is out of the scope of this contribution and, from our perspective, also out of the scope of the Ocean State Report goal.

Therefore, as can be seen in the sentence in L189:

"This result is consistent with Pascual-Collar et al. (2019) that described a displacement of the MOW boundaries towards the west in the Horseshoe basin in the period 2006-2017."

This work only tries to check whether the obtained results are consistent with the available knowledge of the region. Any discussion regarding the validity of previous hypothesis, would be only applicable in case of find contradictory results, which is not the case.

However, as explained in General Comments, we propose to modify the section providing a more detailed explanation of results and its links with Leadbetter et al. (2007), Bozec et al. (2011), and Pascual-Collar et al. (2019).

*l.194. Again about the limits of the MOW, I disagree that the warming in the 43N box (on the northwards pathway of the MOW vein) indicates a westward movement of the MOW tongue.*

We can agree that, in this sentence, there is no causal relationship between warming/cooling and displacement of the MOW tongue. Therefore, we will reword the sentence as follows:

*"The warming of the westward limits of the MOW waters (boxes 34N and 43N) and the cooling of the northward boundary of MOW (subregion 49N) in the period 2005-2020 are consistent with a westward movement of the MOW tongue described by Bozec et al. (2011)."*

*l.202 Section 6. Data availability. The products used have been already described; I do not think this 2-line section is necessary.*

We may agree on this, but again it is a requirement related to the Ocean State Report structure. Further discussion about this point should be done with the OSR editors.

*l.216 OHC changes expressed in W/m3 (power density), should read W/m2?. Also in Table 2.*

OHC is usually presented as an integration from ocean's surface down to a static depth (f.e. 0-150 m, 0-700 m, and 0-2000m), therefore results are expressed in $J \cdot m^{-2}$. However, Table 2 shows OHC changes (usually in $W \cdot m^{-2}$) for three layers with different thickness so, to make them comparable, results have been divided by the layer thickness resulting $W \cdot m^{-3}$.

We will include this information in the text.

*Figure 3. The procedure to obtain the averaged dots (markers) in the TS diagrams is not explained.*

Markers in figure 3b, 3c, and 3d show the spatial average (computed on the corresponding region) of $\theta$ and S at each vertical level. Since these spatial averages are computed for timeseries in annual basis, a mean value ($\theta$ and S) is obtained for each depth, and year.

We agree with the reviewer that this information is not clearly included in the text, so we commit to solve this issue in the next version of the manuscript.

---

## Author Comment (AC2)

***Review of "Ocean Heat Content in the Iberian-Biscay-Ireland regional seas" by PascualCollar et al (2022) for 7th edition of the Copernicus Marine Service Ocean State Report (OSR 7)***

***Summary***

*This work reports on Ocean Heat Content (OHC) trends over the Iberian-Biscay-Ireland (IBI) region based on Copernicus Marine products. Averaged over the entire region, OHC from 0 to 2000m shows a positive trend of 0.5 ± 0.4 W/m2. Regional values display an interesting dipole pattern, with negative trends affecting the offshore ocean north of 38˚N, and a positive 'tongue' between 31˚N and 38˚N. The authors then link the OHC trends with the main water masses in the region, revealing how mainly the Mediterranean Outflow Water impacts the region. As OHC is considered an Ocean Monitoring indicator, linking OHC with water masses is a relevant issue, as it brings insights in the mechanisms driving OHC changes. Hence, the preprint is of scientific value and in the scope of the Ocean State Report. However, there is a divergence between the values reported in Figure 1 and the ones reported in the text. Depending on which value is correct, the interpretation of the results can change a lot, and so the conclusions found in the text. With this issue being solved, and the general remarks below being addressed, I believe this would be a good addition to the Report.*

Authors thank the review and the constructive comments. We consider most of the observations proposed by the reviewer contribute to improving the quality of the paper. Therefore, we are open to write a new version of the manuscript including them. Following are summarized some comments about the main issues observed:

Regarding the error found in Figure 1, the authors would like to apologize for it, and we confirm that it is a mistake on figure edition, so discussion and conclusions reported in text are still pertinent.

The reviewer suggestions on the paper structure and its organization will be considered by the authors to improve the readability of the resulting manuscript.

There are some comments recommending extending the information included in the paper. These observations mainly affect the introduction. We consider most of these recommendations are topics that will improve the background of the manuscript. But, considering the size constrictions of the Ocean State Report contributions, these recommendations will be followed including brief additions.

All the reviewer suggestions will be included in the modified manuscript. However, the authors want to state that such new manuscript will have some major changes, some of them following reviewer RC2 recommendations. Such changes will be mainly focused on:

1. Figure 1 will be modified showing trends in the whole time series (period 1993-2020) and in the period 2005-2019. The text will be consequently adapted to discuss these trends.

2. We will modify section 4 showing the regional trends not only for the period 2005-2019 but also for the whole period 1993-2020. The different results computed in these two periods will be discussed in the text.

3. In section 5 we propose to analyse the OHC profiles in two different periods: 1993-2010 and 2010-2018. These two periods are proposed to represent two different behaviours of NAO index (negative trend in the period 1993-2010 and positive trend

in the period 2010-2018). We consider this analysis can reinforce our conclusions highlighting the concordance of results with the hypothesis developed on Leadbetter et al. (2007), Bozec et al. (2011), and Pascual-Collar et al. (2019).

**General comments:**

1. As mentioned in the summary, there is a divergence between the values reported in Figure 1 and the ones reported in the text: Figure 1 reports a trend of 0.4 ± 0.3 of OHC integrated from surface to 150m and of 0.1 ± 0.1 from surface to 700m, while in the text the authors report the 0.4 ± 0.3 being from 0-700m.

   The authors apologize for this mistake, the panels in Figure 1 are incorrectly positioned. Please, see reply to Specific Comment 15.

2. Structure: The structure of the report could be improved. There are methodological details in the results that belong in Data & Methods Section, and results in the Conclusion (which will be indicated in the specific comments).

   The structure of the contribution will be modified to provide a more organised report. Some observations are given in Specific Comments that we will implement in the text.

   See reply to Specific Comments 6,11, 12, 17, 22, and 28.

3. Data & Methods: The methods section could also benefit from explaining how trends were compute (e.g., ordinary least-squares, auto-regressive models, mann-kendall, …), and by giving some brief details about the data sets used in the analysis. Why were only these 4 datasets used? There are other global products besides GLO-REA, CORA and ARMOR which could have been used (for example, CGLOR and ORAS-5 (GLOBAL_REANALYSIS_PHY_001_031)), why were these ones selected? It is also important to mention if the same source of temperature and salinity data are included in these products, since you present an ensemble mean in the results. It would also be relevant to mention here that GLO-REA is used as boundary condition for IBI-REA (and the possible implications of this for the ensemble mean).

   All this information will be included in the revised manuscript.

4. Discussion: There is a lack of discussion with previous works. For example, in the introduction it is mentioned that previous reports "detected warmings of 0.9±0.4 W/m2 in the upper 700m in the study region". The present study reports a trend of 0.1±0.1 (or 0.4±0.3, depending on which value is correct). The divergence of values should be discussed in the manuscript: how come such smaller trends have been detected now?

   The warmings of 0.9 Wm2 were reported in von Schuckmann et al. (2016) and computed over the period 1993-2015. However, the results reported in this work for the period 2015-2020 (20% of data) show weak or even negative anomalies.

   The authors agree on the reviewer point, the discussion of divergences with previous editions of OSR is due. Therefore, this information will be included in a future version of the paper.

   It worth mentioning that this result is also affected by conclusion of Specific Comment 16.

5. Region map: This is more of a suggestion, but I think the Introduction could benefit from a map showing the IBI region and indicating the main water masses which are discussed in the

text. This would make the text of the introduction read easier, and the interpretation of the results.

Authors understand this suggestion, we will include the description of the work area, however the inclusion of a new figure could collide with the report constraints. Further discussion about this point should be done with the OSR editors.

***Specific comments:***

1.  *L10: I would change 'model' to 'reanalysis', or say 'model reanalysis' (as in L106).*

    Suggestion accepted; it will be included in text.

2.  *The abstract and short summary mention that the work provides an "exhaustive analysis", which made me expect either a lot of data sets included in the analysis, or a lot of results. I think the world "thorough" or another one would be better suited here.*

    The suggestion is accepted, the text will be modified.

3.  *L39: Could you add some examples of Ocean Monitoring Indicators?*

    Suggestion accepted and included in the text. Some other examples of CMEMS OMIs will be mentioned.

4.  *L42: It is said that the product "GLOBAL_OMI_OHC_trend" will hereafter be referred as GLO-OMI-trend, but in the next sections it is referred mainly as "GLOBAL_OMI_OHC_trend" (for example in paragraph 2 of section 4).*

    All references to GLOBAL_OMI_OHC_trend will be changed by GLO-OMI-trend.

5.  *L49: for which period is the trend of 0.9 ± 0.4 W/m2 from? Can you also add reference to the works that reported such trend?*

    This trend was computed for the period 1993-2015. Such information will be included in the text. The paragraph will be modified presenting the referenced works next to this sentence.

6.  *The paragraph about water masses (L60-67) in the study region should come before the paragraph describing how previous studies linked these water masses with OHC (L49-59). This would be a good place to actually introduce the study region.*

    We will carefully consider this suggestion when drafting the new version of introduction.

7.  *L68-70: Are these results also for the IBI region? This should be mentioned. And I guess this example belongs to the paragraph from L49-59.*

    We will carefully consider this suggestion when drafting the new version of introduction.

8.  *L72: You say MOW does not dominate the changes in the Mediterranean Sea Water… could you mention what does then?*

    Changes are mainly dominated by year-to-year processes. This paragraph will be rewritten to avoid misunderstandings.

9.  *L81-83: Here the objective of the work is described. I think should also mention for which periods OHC trends will be analyzed.*

    Suggestion accepted; this information will be included.

10. *L102: You mention "a collection of data sources", do you mean several Copernicus products (such as CORA, Glorys, …) or several sources of temperature and salinity data (such as Argo gloats, CTDs, XBTs, …)? It was not clear for me. Also "several data sources" (L101) refers to which? If it's to the products in table 4, I don't think 4 products can be classified as 'several'.*

    We will clarify this referencing the products listed in Table 1.

11. *L114: Here should be mentioned which grid resolution was selected for the ensemble mean.*

   The resolution used for the ensemble depends on whether the result comprises lat/lon information:

   In the case of the time series show in Figure 1. Each product preserves its native resolution, since the time series are computed as spatial averages, each product provides a time series computed over its own native grid. Then, the ensemble averages the time series provided for all products.

   In results that comprises lat/lon information (i.e. the ones presented in Figure 2) all products are reprojected over the IBI-REA mesh, as explained in Section 4.

   We understand that according to other suggestions (Specific Comments 12 and 17), this paragraph must be rewritten providing all the information regarding the averaging of all products in methods section.

12. *L116-119: this should be in the methods section.*

   This information will be moved to methods. See reply to Specific Comment 11.

13. *Analysis of OHC timeseries: I was wondering if instead of only integrating the values from the surface until the reference depth (150, 700 and 2000m), the analysis could benefit from integrating in intervals? From 0-150m, 150-700m, 700-2000m. This way the actual contribution from each layer would be clearer. With always integrating from the surface, the contribution and uncertainties become cumulative, and not independent, so the results from 0-2000m should reflect the behavior from 0-700m, in addition to the contribution of 700-2000m.*

   One of the main objectives of this paper is to examine the interpretation of results integrated from 0-150, 0-700, and 0-2000. This is because we pursue to analyse the sensitivity of this standardised parameters to its vertical variability. With this, the work only provide information about the IBI region; it also provide results that must be considered when interpreting OHC integrated in these typical layers (even when are obtained outside the IBI region).

14. *L123-L127: not only this change in the uncertainties is 'remarkable'. I also found very interesting how the uncertainties are much wider from 99-2002. Could you comment on what could be the cause of that? Would it be related to climate fluctuations, or also to the quality (and quantity) of data?*

   This is an interesting appreciation; we will discuss this result internally and we will comment it in the text. Our fist interpretation is related with the progressive development of the Argo array that started in 2000.

15. *L128-L131 & Figure 1: Here is where the results of Figure 1 and the text do not match. The text says the trend of 0.4±0.2 is for the upper 700m, which are in the figure the numbers for the upper 150m. Then it is concluded that "such a trend does not affect the mixing layer. This result suggests a dominance of the variability at intermediate-deep-levels over the OHC trend". Considering the mixing layer is within the upper 150m (as stated in L100, but could actually be reinforced here). If the numbers in Figure 1 are correct, then I would interpret as: the upper 150m have a positive trend; the upper 700m have an insignificant trend,*

*dominating the variability of OHC, instead of the trend; the overall integrated OHC in the upper 2000m is a positive trend.*

We apologise for this mistake, the labelling of panels in Figure 1 is wrong, so the figures in panels A and B are switched. The correct figure is here shown:

This result has been double checked and we can confirm that is correct. Every figure is internally created with its own automatic tittle, here shown but hidden in Figure 1 in the paper.

Additionally, this result is more consistent showing an increase of uncertainties with the integration depth.

[Figure]

16. *L131: "It can be appreciated a change of trend after the year 2006". What does this mean exactly? Do you mean that if we would analyse trends from 1993-2006, then we would see a negative one and a positive one for 2006-2020? Or do you mean an acceleration in the rate…? Could this be quantified (by a breakpoint analysis for example)?*

The change of the observed trends linked to the period considered is one of the main points of discussion of this work. Thus, we propose to make this analysis not only in section 3 but also in sections 4 and 5.

Regarding the section 3, the analysis of trends in Figure 1 will be performed for two periods, the whole time series 1993-2020 and the period 2005-2019. The differences will be consequently discussed in the text.

Additionally, as described in general comments, section 4 will show and discuss the regional trends for the periods 2005-2019 and the period 1993-2020. The different results obtained in these two periods will be discussed in the text. In section 5 we propose to analyse the OHC profiles in two different periods: 1993-2010 and 2010-2018. These two periods are proposed to represent two different behaviours of NAO index (negative trend in the period 1993-2010 and positive trend in the period 2010-2018).

17. *L135-138: this should be in the methods section.*

This information will be moved to methods. See reply to Specific Comment 11.

18. *L138: The IBI-REA grid was used as a reference, which is the one with the higher spatial resolution. This mean that CORA and ARMOR had to be downscaled to match this grid resolution, meaning than that each grid cell is not independent anymore (1 value was split*

*into more). Could you comment on effect of this for your analysis? Would your results change if instead of matching IBI-REA resolution, you would match the data set with coarser resolution?*

For this result the OHC is computed from every product using its own grid, trends are then computed over the native grid of the product and finally, OHC trends are interpolated (using bilinear interpolation) to project all results over the same grid. We have not observed mathematical artifacts on this process, so authors do not expect impact on results due to the interpolation.

As the reviewer suggests, every grid point is spited into more, but since the OHC is a parameter averaged on surface units, the splitting of data does not have impact because results are divided by the horizontal surface ($m^2$) of the cell.

The use of a coarser resolution grid would lead to a degradation of IBI-REA and GLO-REA results. We consider that this procedure would not have any advantage that would justify such a degradation.

The text will be modified including a deeper explanation of the interpolation procedure done on this step.

19. *L144: what do you mean by "inhomogeneity of uncertainties"?*

We refer to the statistical meaning of the word inhomogeneity, a lack of uniformity. With this expression we refer to the decrease of uncertainties seen after 2003 in time series. This variability of uncertainties is previously explained on section 3 (L123).

The authors understand that this sentence is too summarized and can be interpreted in different ways. To avoid misunderstandings, the sentence will be rewritten including more information.

20. *L148-152: You talk here about some of the divergencies of the ensemble and the OMI product. Might also be worth mentioning that positive trends are seen in the ENS in along the Northwestern European shelf, but this is not seen in the OMI product.*

We thank for this suggestion; we consider it is an interesting result not properly commented in text.

21. *Another thing that caught my eye in Figure 2 is between 30-36°N the trends are not significant for 0-150 and 0-700m, but they are from 0-2000m. Suggesting that the uncertainties become smaller when the deeper layers are added, which seems counterintuitively to me. Or maybe is just an artifact of the intensification of the trends with the deeper layers (as stated in L155)? Could you comment on that?*

This is mainly an artifact that authors associate with the static integration depths. It must be considered that, since the OHC is an integrated variable, results show a summation of all layers weighted by the layer thickness.

We can explain these results observing the T/S diagrams in Figure 3d:

- The layer 0-150 does not show trends, therefore results are not significant.

- The layer 0-700 mixes results from the upper layer (not significant trends), with significant cooling of the underlying levels, and the significant warming of the deepest layers at 600-700m.

Additionally, we should also consider the decrease of interannual variability with depth.

22. *L162-L165: this should be in the methods section.*

After a carefully read, authors do not detect any methods explained in lines 162-165. We understand the reviewer could refer to lines 172-175. This information will be moved to methods. See reply to Specific Comment 11.

23. *L175: It was unclear to me what was meant by 'yearly basis'. Did you use yearly values for the diagrams? Looking at Figure 3 and the results, I believe you used trends of temperature and salinity for the diagrams... but this was unclear.*

Markers in figure 3b, 3c, and 3d show the spatial average (computed on the corresponding region) of θ and S at each vertical level. Since these spatial averages are computed for timeseries in annual basis, a mean value (θ and S) is obtained for each depth, and year.

We understand the reviewer's comment and we will clarify this sentence.

24. *Figure 3: Just a suggestion: since you discuss the water masses, maybe you could actually add boxes in the T/S diagrams indicating where is each water mass? (I had to go back and forth between the definition of the water masses and the figures to know which one was at the surface and which one was the intermediate and the bottom one.*

This suggestion is easy to implement and will make the reading easier.

25. *L185: "a positive trend of temperature entails a positive trend of salinity". Is it only temperature influencing salinity, or vice-versa?*

There is a correlation between temperature and salinity, thus both variables are linked and variation in one of them are followed by variation in the other one.

26. *L198: you state that the positive trends in LSW could be explained by the connection between the MOV and LSW. But what about box 49N, in which MOW has negative trends? Then the positive trends seen in LSW in that box could not be due to the MOW influence…*

Authors understand the doubts of the reviewer on this sentence. We will elaborate this conclusion, or we will remove it.

27. *L209-210 and L218: these conclusions are wrong if the values in Figure 1 of 0-150m are correct.*

Please, see Specific Comment 15.

28. *L220-228: This is more of results than conclusion. Maybe would fit better in the previous section.*

Since one of the main objectives of OSR is to "provide a comprehensive and state-of-the art assessment of the state of the ocean" we consider the rate of warming/cooling of water masses a relevant result that must be highlighted in the conclusions. However, we understand the suggestion and we accept that all references to previous works fits better as a discussion of results, they will be removed from this section.

***Technical corrections:***

We thank all corrections, all of them will be included.

- *L44: Here is the first time the acronym "IBI" is used in the main text. You should write it out Iberian-Biscay-Ireland the first time it is mentioned.*
- *L45: Typo: 'info' -> information*
- *L46: Typo: 'regions closed to coastal areas -> close*
- *L49: OSRs is mentioned for the first time, without being defined (and appears written out in L91). I suggest just using the expanded version in both places, since the texts already has a lot of acronyms.*
- *L54 and L63 both define Mediterranean Outflow Water (MOW). An acronym is usually only defined once.*
- *L57-59: You should either use 'although' or 'nevertheless', but not both in the same sentence. If you keep 'although', then the semi-column after limited should be replace by a period. And in the last part of the sentence "these studies give some…" you can remove "these studies" and change "give" to "giving".*
- *L80: NAO hasn't been defined. And it actually appears written out in L218.Again, I suggest just using the expanded version in both places, since the texts already has a lot of acronyms.*
- *L103-105: "Since the objective of this work is the analysis of OHC in the IBI region integrating results from surface down to a maximum of 2000 m depth., Tthis study has included all Copernicus Marine products that provide gridded data of potential temperature with vertical coverage from surface down to at least 2000 m. '*
- *L146: typo: booth -> both*
- *L155: It should be 'larger' instead of 'bigger': "the larger the integration depth, the larger is the observed trend".*
- *L208: "being these indicators useful …" -> "these indicators are useful …"*
- *L210: "Despite the regional analysis …" -> "While the regional analysis …"*
- *L213: end of the line, between OHC and Table 2* be a period and not a comma.

---

## Author Response (AR2)

Dear editor;

We thank the valuable comments given in the review. The authors agree with all of them, and we have modified the manuscript including all recommendations.

Additionally, some changes related with the first table (product table) have been included to properly reference the data used.

Following we include a brief point-by-point reply to the comments:

L54 – 'proposed covering' – something missing
   Accepted: "are proposed to cover…"

L58 – remove 'constitute'
   Accepted

L68 – remove comma after dynamics
   Accepted

L70 – in 'the' IBI region
   Accepted

L135: 'differentiation' implies a mathematical derivative function. I think you actually wanted to say 'difference'?
   Accepted

L136 – the different products were interpolated to which grid?
   The selected grid to homogenise de spatial information is explained few lines later. (141-145).

L132-145: not necessary to say lat/lon maps, just maps is enough; The same for lat/lon ensemble, 'spatial ensemble' is enough.
   Accepted

L139: "Then the vertical integrations and spatial average 'were' (not was) obtained"
   Accepted

L140-141 says the same thing as L136-137.
   The first sentence (L136-137) was removed.

L149 – should it be 'Section 5' here? (Section 6 should be the data availability section, which is missing from the revised manuscript).
   Accepted.

L150: Which NAO index was used? The one provided by NOAA[1] and by NCAR [2], for example, are slightly different. And can you add a plot of the NAO index to the manuscript or to supplementary? I am not sure if I would say that it was moderately positive from 1993-2010, but actually mainly negative, with a minimum in 2010, and then a mainly positive behavior. The description in L211-212 seems more accurate.
   This paragraph has been modified avoiding the use of "significant trends" and describing the two periods as representative of NAO change of phase.
   The NOAA web page has been properly referenced in the manuscript.
   A figure of NAO index in annual means has been included as supplementary material.

L156: you use 'time series' here, with space, and in other instances 'timeseries'. Please be consistent (I would suggest time series, with space).
   Accepted

L167: And are the trends from 0-2000m significantly? A trend of 0.4 with an uncertainty of 0.39 doesn't seem very robust… I would still dare to say that the intermediate layers (150-

700m) dominate the OHC variability, since the difference from 0-700 and 0-2000m is almost negligible.

We agree that this result is in the limit of significance, thus we have considered it as a mathematical result with no real significance.

Some changes have been included in the paragraph and in the first paragraph of conclusions.

L190: typo 'shake'

Accepted

L192: Suggest to add a reference depth to the mixing layer, since this can vary a lot depending on the region.

To avoid confusion, we have changed the term by upper layer.

L223: at the end of the sentence, I think it should be '35' and not '24', referring to the region centered at 35°N.

Accepted

L241: 'boundary transports of MOW and NAO'. I guess you meant to say something else than 'NAO' here? The boundary of MOW should either be with other water masses or with another region.

This sentence has been corrected: "significant anti-correlation between the westward and northward transport of MOW"

L246: 'occupies' should then refer to a depth; a water mass occupy a certain depth, or has a certain thickness.

This sentence has been corrected: "Considering that each water mass has different thickness".

L255: remove comma after both.

Accepted

L259: Suggest to replace 'despite' for 'although'.

Accepted

L251: Section 6 is missing.

Corrected

---

## Author Response (AR3)

Dear editor,

We apologize for the error in the figure numbering and kindly ask you to accept this new version of the manuscript.

Best regards,

Álvaro de Pascual Collar